# Cross-lingual Retrieval for Iterative Self-Supervised Training

**Chau Tran**
Facebook AI
chau@fb.com

**Yuqing Tang**
Facebook AI
yuqtang@fb.com

**Xian Li**
Facebook AI
xianl@fb.com

**Jiatao Gu**
Facebook AI
jgu@fb.com

## Abstract

Recent studies have demonstrated the cross-lingual alignment ability of multilingual pretrained language models. In this work, we found that the cross-lingual alignment can be further improved by training seq2seq models on sentence pairs mined using their own encoder outputs. We utilized these findings to develop a new approach — cross-lingual retrieval for iterative self-supervised training (CRISS), where mining and training processes are applied iteratively, improving cross-lingual alignment and translation ability at the same time. Using this method, we achieved state-of-the-art unsupervised machine translation results on 9 language directions with an average improvement of 2.4 BLEU, and on the Tatoeba sentence retrieval task in the XTREME benchmark on 16 languages with an average improvement of 21.5% in absolute accuracy. Furthermore, CRISS also brings an additional 1.8 BLEU improvement on average compared to mBART, when finetuned on supervised machine translation downstream tasks. Our code and pretrained models are publicly available. [1]

## 1  Introduction

Pretraining has demonstrated success in various natural language processing (NLP) tasks. In particular, self-supervised pretraining can learn useful representations by training with pretext task, such as cloze and masked language modeling, denoising autoencoder, etc. on large amounts of unlabelled data [11, 25, 28, 32, 35, 51]. Such learned "universal representations" can be finetuned on task-specific training data to achieve good performance on downstream tasks.

More recently, new pretraining techniques have been developed in the multilingual settings, pushing the state-of-the-art on cross-lingual understandin, and machine translation. Since the access to labeled parallel data is very limited, especially for low resource languages, better pretraining techniques that utilizes unlabeled data is the key to unlock better machine translation performances [9, 27, 42].

In this work, we propose a novel self-supervised pretraining method for multilingual sequence generation: Cross-lingual Retrieval for Iterative Self-Supervised training (CRISS). CRISS is developed based on the finding that the encoder outputs of multilingual denoising autoencoder can be used as language agnostic representation to retrieve parallel sentence pairs, and training the model on these retrieved sentence pairs can further improve its sentence retrieval and translation capabilities in an iterative manner. Using only unlabeled data from many different languages, CRISS iteratively mines

for parallel sentences across languages, trains a new better multilingual model using these mined sentence pairs, mines again for better parallel sentences, and repeats.

In summary, we present the following contributions in this paper:

- We present empirical results that show the encoder outputs of multilingual denoising autoencoder (mBART) represent language-agnostic semantic meaning.

- We present empirical results that show finetuning mBART on only one pair of parallel bi-text will improve cross-lingual alignment for all language directions.

- We introduce a new iterative self-supervised learning method that combines mining and multilingual training to improve both tasks after each iteration.

- We significantly outperform the previous state of the art on unsupervised machine translation and sentence retrieval.

- We show that our pretraining method can further improve the performance of supervised machine translation task compared to mBART.

This paper is organized as follows. Section 2 is devoted to related work. Section 3 introduces improvable language agnostic representation emerging from pretraining. Section 4 describes the details of cross-lingual retrieval for iterative self-supervised training (CRISS). Section 5 evaluates CRISS with unsupervised and supervised machine translation tasks as well as sentence retrieval tasks. Section 6 iterates over ablation studies to understand the right configurations of the approach. Then we conclude by Section 7.

## 2 Related work

**Emergent Cross-Lingual Alignment**   On the cross-lingual alignment from pretrained language models, [49] [33] present empirical evidences that there exists cross-lingual alignment structure in the encoder, which is trained with multiple languages on a shared masked language modeling task. Analysis from [46] shows that shared subword vocabulary has negligible effect, while model depth matters more for cross-lingual transferability. In English language modeling, retrieval-based data augmentation has been explored by [20] and [15]. Our work combines this idea with the emergent cross-lingual alignment to retrieve sentences in another language instead of retrieving paraphrases in the same language in unsupervised manner.

**Cross-Lingual Representations**   Various previous works have explored leveraging cross-lingual word representations to build parallel dictionaries and phrase tables, then applying them to downstream tasks [1, 2, 3, 4, 24, 29]. Our work shows that we can work directly with sentence-level representations to mine for parallel sentence pairs. Additionally, our approach shares the same neural networks architecture for pretraining and downstream tasks, making it easier to finetune for downstream tasks such as mining and translation.

There is also a large area of research in using sentence-level representations to mine pseudo-parallel sentence pairs [6, 8, 14, 17, 39, 40, 43]. Compared to supervised approaches such as [14, 40], CRISS performs mining with unsupervised sentence representations pretrained from large monolingual data. This enables us to achieve good sentence retrieval performance on very low resource languages such as Kazakh, Nepali, Sinhala, Gujarati. Compared to [17], we used full sentence representations instead of segment detection through unsupervised word representations. This enables us to get stronger machine translation results.

**Multilingual Pretraining Methods**   With large amounts of unlabeled data, various self-supervised pretraining approaches have been proposed to initialize models or parts of the models for downstream tasks (e.g. machine translation, classification, inference and so on) [11, 12, 25, 27, 28, 32, 35, 36, 37, 42, 51]. Recently these approaches have been extended from single language training to cross-lingual training [9, 22, 27, 45]. In the supervised machine learning literature, data augmentation [5, 21, 41, 50] has been applied to improve learning performance. To the best our knowledge, little work has been explored on self-supervised data augmentation for pretraining. This work pretrains multilingual model with self-supervised data augmenting procedure using the power of emergent cross-lingual representation alignment discovered by the model itself in an iterative manner.

**Unsupervised Machine Translation**  Several authors have explored unsupervised machine translation techniques to utilize monolingual data for machine translation. A major line of work that does not use any labeled parallel data typically works as follow: They first train an initial language model using a noisy reconstruction objective, then finetune it using on-the-fly backtranslation loss [13, 23, 26, 27, 42]. Our work differs from this line of work in that we do not use backtranslation but instead retrieve the target sentence directly from a monolingual corpus. More recently, [38] and [48] start incorporating explicit cross-lingual data into pretraining. However, since the quality of cross-lingual data extracted is low, additional mechanisms such as sentence editing, or finetuning with iterative backtranslation is needed. To the best of our knowledge, our approach is the first one that achieves competitive unsupervised machine translation results without using backtranslation.

## 3  Self-Improvable Language Agnostic Representation from Pretraining

We start our investigation with the language agnostic representation emerging from mBART pretrained models [27]. Our approach is grounded by the following properties that we discovered in mBART models: (1) the mBART encoder output represents the semantics of sentences, (2) the representation is language agnostic, and (3) the language agnostics can be improved by finetuning the mBART models with bitext data of a small number of language pairs (or even only 1 pair of languages) in an *iterative manner*. We explain these findings in details in this section and the next section on the iterative procedure.

### 3.1  Cross-lingual Language Representations

We use mBART [27] seq2seq pre-training scheme to initialize cross-lingual models for both parallel data mining and multilingual machine translation. The mBART training covers $N$ languages: $\mathcal{D} = \{\mathcal{D}_1, ..., \mathcal{D}_N\}$ where each $\mathcal{D}_i$ is a collection of monolingual documents in language $i$. mBART trains a seq2seq model to predict the original text $X$ given $g(X)$ where $g$ is a noising function, defined below, that corrupts text. Formally we aim to maximize $\mathcal{L}_\theta$:

$$\mathcal{L}_\theta = \sum_{\mathcal{D}_i \in \mathcal{D}} \sum_{x \in \mathcal{D}_i} \log P(x|g(x); \theta) , \tag{1}$$

where $x$ is an instance in language $i$ and the distribution $P$ is defined by the Seq2Seq model. In this paper we used the mbart.cc25 checkpoint [27] open sourced in the Fairseq library [30] [2]. This model is pretrained using two types of noise in $g$ — random span masking and order permutation — as described in [27].

With a pretrained mBART model, sentences are then encoded simply by extracting L2-normalized average-pooled encoder outputs.

### 3.2  Case Study

To understand the language agnosticity of the mBART sentence representations, we study sentence retrieval tasks. For each language pair, we go through each sentence in the source language, find the closest sentence to that sentence in the target language (using cosine similarity), and report the average top-1 retrieval accuracy for each language pair. We use the TED58 dataset which contains multi-way translations of TED talks in 58 languages [34][3].

The sentence retrieval accuracy on this TED dataset is depicted in Figure 1. The average retrieval accuracy is 57% from the mBART model which is purely trained on monolingual data of 25 languages without any parallel data or dictionary; the baseline accuracy for random guessing is 0.04%. We also see high retrieval accuracy for language pairs with very different token distribution such as Russian-German (72%) or Korean-Romanian (58%). The high retrieval accuracy suggests that mBART model trained by monolingual data of multiple languages is able to generate language agnostic representation that are aligned at the semantic level in the vector space.

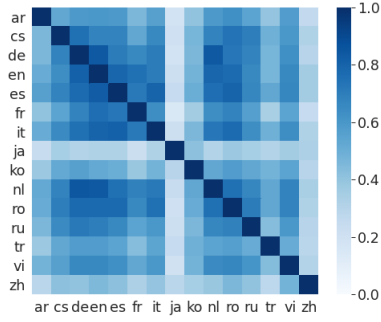

Figure 1: Sentence retrieval accuracy using encoder outputs of mBART

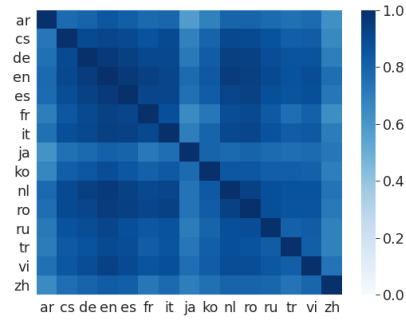

Figure 2: Sentence retrieval accuracy using encoder outputs of mBART finetuned on Japanese-English parallel data

Moreover the cross-lingual semantic alignment over multiple languages not only emerges from the monolingual training but also can be improved by a relatively small amount of parallel data of just one direction. Figure 2 shows the sentence retrieval accuracy of an mBART model that are finetuned with Japanese-English in the IWSLT17 parallel dataset ($223,000$ sentence pairs) [7]. Even with parallel data of one direction, the retrieval accuracy improved for all language directions by 27% (absolute) on average.

Inspired by these case studies, we hypothesize that the language agnostic representation of the pretrained model can be self-improved by parallel data mined by the model itself without any supervised parallel data. We will devote section 4 to details of the self-mining capability and the derived Cross-lingual Retrieval Iterative Self-Supervised Training (CRISS) procedure.

## 4 Cross-lingual Retrieval for Iterative Self-Supervised Training (CRISS)

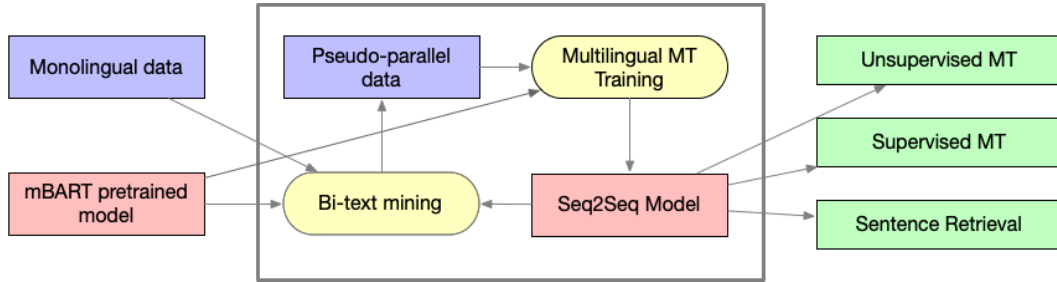

Figure 3: Overview of Cross-lingual retrieval self-supervised training

---

**Algorithm 1** Unsupervised Parallel Data Mining

---

1: **function** MINE($\Theta, D_i, D_j$)
2:     **Input:** (1) monolingual data sets $D_i$ and $D_j$ for language $i$ and $j$ respectively, (2) a pretrained model $\Theta$,
3:     Set $k$, $M$, $\tau$ to be the desired KNN size, the desired mining size, and the desired minimum score threshold respectively
4:     **for** each $x$ in $D_i$, each $y$ in $D_j$ **do**
5:         $x, y \leftarrow \text{Embed}(\Theta, x), \text{Embed}(\Theta, y)$
6:         $N_x, N_y \leftarrow KNN(x, D_j, k), KNN(y, D_i, k)$         ▷ Using FAISS [19]
7:     **end for**
8:     **return** $D' = \{(x,y)\}$ where $(x,y)$ are the top $M$ pairs s.t. score$(x,y) \geq \tau$ following Equation 2
9: **end function**

---

---

**Algorithm 2** CRISS training

---
1: **Input:** (1) monolingual data from $N$ languages $\{D_n\}_{n=1}^{N}$, (2) a pretrained mBART model $\Psi$, (3) total number of iterations $T$
2: Initialize $\Theta \leftarrow \Psi$, $t = 0$
3: **while** $t < T$ **do**
4:     **for** every language pairs $(i, j)$ where $i \neq j$ **do**
5:         $D'_{i,j} \leftarrow Mine(\Theta, D_i, D_j)$                                    ▷ Algorithm 1
6:     **end for**
7:     $\Theta \leftarrow MultilingualTrain(\Psi, \{D'_{i,j} \mid i \neq j\})$ ▷ Note: Train from the initial mBART model $\Psi$
8: **end while**

---

In this section, we make use the of the language agnostic representation from mBART described in Section 3 to mine parallel data from a collection of monolingual data. The mined parallel data is then used to further improve the cross-lingual alignment performance of the pretrained model. The mining and improving processes are repeated multiple times to improve the performance of retrieval, unsupervised and supervised multilingual translation tasks as depicted in Figure 3 using Algorithm 2.

**Unsupervised parallel data mining** To mine parallel data without supervised signals, we employ the margin function formulation [5, 6, 40] based on $K$ nearest neighbors (KNN) to score and rank pairs of sentences from any two languages. Let $x$ and $y$ be the vector representation of two sentences in languages $i$ and $j$ respectively, we score the semantic similarity of $x$ and $y$ using a ratio margin function [5] defined as the following:

$$\text{score}(x, y) = \frac{\cos(x, y)}{\sum_{z \in N_x} \frac{\cos(x, z)}{2k} + \sum_{z \in N_y} \frac{\cos(z, y)}{2k}} \tag{2}$$

where $N_x$ is the KNN neighborhood of $x$ in the monolingual dataset of $y$'s language; and $N_y$ is the KNN neighborhood of $y$ in the monolingual dataset $x$'s language. The margin scoring function can be interpreted as a cosine score normalized by average distances (broadly defined) to the margin regions established by the cross-lingual KNN neighborhoods of the source and target sentences respectively. The KNN distance metrics are defined by $\cos(x, y)$. We use FAISS [19] — a distributed dense vector similarity search library — to simultaneously search for all neighborhoods in an efficient manner at billion-scale. Thus we obtain the mining Algorithm 1.

We find that in order to train a multilingual model that can translate between $N$ languages, we don't need to mine the training data for all $N * (N - 1)$ language pairs, but only a subset of them. This can be explained by our earlier finding that finetuning on one pair of parallel data helps improve the cross-lingual alignment between every language pair.

**Iteratively mining and multilingual training** Building on the unsupervised mining capability of the pretrained model (Algorithm 1), we can conduct iteratively mining and multilingual training procedure (Algorithm 2) to improve the pretrained models for both mining and downstream tasks. In Algorithm 2, we repeat mining and multilingual training $T$ times. On multilingual training, we simply augment each mined pair $(x, y)$ of sentences by adding a target language token at the beginning of $y$ to form a target language token augmented pair $(x, y')$. We then aggregate all mined pairs $\{(x, y)'\}$ of the mined language pairs into a single data set to train a standard seq2seq machine translation transformer models [44] from the pretrained mBART model. To avoid overfitting to the noisy mined data, at each iteration we always refine from the original monolingual trained mBART model but only update the mined dataset using the improved model (line 7 of Algorithm 2).

## 5 Experiment Evaluation

We pretrained an mBART model with Common Crawl dataset constrained to the 25 languages as in [27] for which we have evaluation data. We also employ the same model architecture, the same BPE pre-processing and adding the same set of language tokens as in [27]. We keep the same BPE vocab and the same model architecture throughout pretraining and downstream tasks.

On mining monolingual data, we use the same text extraction process as described in [40] to get the monolingual data to mine from, which is a curated version of the Common Crawl corpus. We

refer the reader to [40, 47] for a detailed description of how the monolingual data are preprocessed. For faster mining, we subsample the resulting common crawl data to 100 million sentences in each language. For low resources, we may have fewer than 100 million monolingual sentences. The statistics of monolingual data used are included in the supplementary materials.

We set the $K = 5$ for the KNN neighborhood retrieval for the margin score functions (Equation 2). In each iteration, we tune the margin score threshold based on validation BLEU on a sampled validation set of size 2000. The sizes of mined bi-text in each iteration are included in the supplementary materials.

Our default configuration mines sentences to and from English, Hindi, Spanish, and Chinese, for a total of 90 languages pairs (180 language directions) instead of all 300 language pairs (600 directions) for the 25 languages.

With the mined 180 directions parallel data, we then train the multilingual transformer model for maximum $20,000$ steps using label-smoothed cross-entropy loss as described in Algorithm 2. We sweep for the best maximum learning rate using validation BLEUs.

After pretraining, the same model are evaluated with three tasks: sentence retrieval, unsupervised machine translation, and supervised machine translation tasks. For supervised machine translation, we use CRISS model to initialize the weights to train models for supervised machine translation.

## 5.1 Unsupervised Machine Translation

We evaluate CRISS on unsupervised neural machine translation benchmarks that cover both low resource and high resource language directions. For English-French we use WMT'14, for English-German and English-Romanian we use WMT'16 test data, and for English-Nepali and English-Sinhala we use Flores test set [16]. For decoding in both the unsupervised and supervised machine translation tasks, we use beam-search with beam size 5, and report the final results in BLEU [31]. To be consistent with previous literature, we used multi-bleu.pl[4] for evaluation.

As shown in Table 1, on these unsupervised benchmarks the CRISS model overperforms state-of-the-art in 9 out of 10 language directions. Our approach works well on dissimilar language pairs, achieving $14.4$ BLEU on Nepali-English (improving $4.4$ BLEU compared to previous method), and $13.6$ BLEU on Sinhala-English (improving $5.4$ BLEU compared to previous method). On similar language pairs, we also improved Romanian-English from $33.6$ to $37.6$ BLEU, German-English from $35.5$ to $37.1$ BLEU. We also report translation quality for other language pairs that do not have previous benchmark in the supplementary materials.

| Direction | en-de | de-en | en-fr | fr-en | en-ne | ne-en | en-ro | ro-en | en-si | si-en |
|---|---|---|---|---|---|---|---|---|---|---|
| CMLM [38] | 27.9 | 35.5 | 34.9 | 34.8 | - | - | 34.7 | 33.6 | - | - |
| XLM [10] | 27.0 | 34.3 | 33.4 | 33.0 | 0.1 | 0.5 | 33.3 | 31.8 | 0.1 | 0.1 |
| MASS [42] | 28.3 | 35.2 | 37.5 | 34.9 | - | - | 35.2 | 33.1 | - | - |
| D2GPO [26] | 28.4 | 35.6 | 37.9 | 34.9 | - | - | **36.3** | 33.4 | - | - |
| mBART [27] | 29.8 | 34 | - | - | 4.4 | 10.0 | 35.0 | 30.5 | 3.9 | 8.2 |
| CRISS Iter 1 | 21.6 | 28.0 | 27.0 | 29.0 | 2.6 | 6.7 | 24.9 | 27.9 | 1.9 | 6 |
| CRISS Iter 2 | 30.8 | 36.6 | 37.3 | 36.2 | 4.2 | 12.0 | 34.1 | 36.5 | 5.2 | 12.9 |
| CRISS Iter 3 | **32.1** | **37.1** | **38.3** | **36.3** | **5.5** | **14.5** | 35.1 | **37.6** | **6.0** | **14.5** |

Table 1: Unsupervised machine translation. CRISS outperforms other unsupervised methods in 9 out of 10 directions. Results on mBART supervised finetuning listed for reference.

## 5.2 Tatoeba: Similarity Retrieval

We use the Tatoeba dataset [6] to evaluate the cross-lingual alignment quality of CRISS model following the evaluation procedure specified in the XTREME benchmark [18].

As shown in Table 2, compared to other pretraining approaches that don't use parallel data, CRISS outperforms state-of-the-art by a large margin, improving all 16 languages by an average of 21.5% in

| Language | ar | de | es | et | fi | fr | hi | it |
|---|---|---|---|---|---|---|---|---|
| XLMR [9] | 47.5 | 88.8 | 75.7 | 52.2 | 71.6 | 73.7 | 72.2 | 68.3 |
| mBART [27] | 39 | 86.8 | 70.4 | 52.7 | 63.5 | 70.4 | 44 | 68.6 |
| CRISS Iter 1 | 72 | 97.5 | 92.9 | 85.6 | 88.9 | 89.1 | 86.8 | 88.7 |
| CRISS Iter 2 | 76.4 | **98.4** | 95.4 | **90** | 92.2 | 91.8 | 91.3 | 91.9 |
| CRISS Iter 3 | **78.0** | 98.0 | **96.3** | 89.7 | **92.6** | **92.7** | **92.2** | **92.5** |
| LASER (supervised) [6] | 92.2 | 99 | 97.9 | 96.6 | 96.3 | 95.7 | 95.2 | 95.2 |

| Language | ja | kk | ko | nl | ru | tr | vi | zh |
|---|---|---|---|---|---|---|---|---|
| XLMR [9] | 60.6 | 48.5 | 61.4 | 80.8 | 74.1 | 65.7 | 74.7 | 68.3 (71.6) |
| mBART [27] | 24.9 | 35.1 | 42.1 | 80 | 68.4 | 51.2 | 63.9 | 14.8 |
| CRISS Iter 1 | 76.8 | 67.7 | 77.4 | 91.5 | 89.9 | 86.9 | 89.9 | 69 |
| CRISS Iter 2 | **84.8** | 74.6 | **81.6** | 92.8 | **90.9** | 92 | 92.5 | 81 |
| CRISS Iter 3 | 84.6 | **77.9** | 81.0 | **93.4** | 90.3 | **92.9** | **92.8** | **85.6** |
| LASER (supervised) [6] | 94.6 | 17.39 | 88.5 | 95.7 | 94.1 | 97.4 | 97 | 95 |

Table 2: Sentence retrieval accuracy on Tatoeba; XLMR results are the previous SOTA (except zh where mBERT is the SOTA). LASER is a supervised method listed for reference

absolute accuracy. Our approach even beats the state-of-the-art supervised approach [6] in Kazakh, improving accuracy from 17.39% to 77.9%. This shows the potential of our work to improve translation for language pairs with little labeled parallel training data.

### 5.3 Supervised Machine Translation

For the supervised machine translation task, we use the same benchmark data as in mBART [27]. We finetune models learned from CRISS iteration 1, 2, and 3 on supervised training data of each bilingual direction. For all directions, we use $0.3$ dropout rate, $0.2$ label smoothing, $2500$ learning rate warm-up steps, $3e-5$ maximum learning rate. We use a maximum of 40K training steps, and final models are selected based on best valid loss. As shown in Table 3, CRISS improved upon mBART on low resource directions such as Gujarati-English (17.7 BLEU improvement), Kazakh-English (5.9 BLEU improvement), Nepali-English (4.5 BLEU improvement). Overall, we improved 26 out of 34 directions, with an average improvement of 1.8 BLEU.

## 6 Ablation Studies

We conduct ablation studies to understand the key ingredients of our methods: the pretrained language model, the performance implications of bilingual training versus multilingual training, and the number of pivot languages used to mine parallel data.

### 6.1 Starting from bilingual pretrained models

To study the benefits of the iterative mining-training procedure on a single language pair (ignoring the effects of multilingual data), we run the CRISS procedure on the English-Romanian language pair. We start with the mBART02 checkpoint trained on English-Romanian monolingual data [27], and apply the CRISS procedure for two iterations. As shown in Table 4 and 5, the CRISS procedure does work on a single language pair, improving both unsupervised machine translation quality, and sentence retrieval accuracy over time. Moreover, the sentence retrieval accuracy on bilingual CRISS-EnRo is higher than that of CRISS25, but the unsupervised machine translation results for CRISS25 are higher. We hypothesize that CRISS-EnRo can mine higher quality English-Romanian pseudo-parallel data, since the the encoder can specialize in representing these two languages. However, CRISS25 can utilize more pseudo-parallel data from other directions, which help it achieve better machine translation results.

| Direction | en-ar | en-et | en-fi | en-gu | en-it | en-ja | en-kk | en-ko | en-lt |
|---|---|---|---|---|---|---|---|---|---|
| mBART | 21.6 | **21.4** | 22.3 | 0.1 | 34 | 19.1 | 2.5 | **22.6** | 15.3 |
| CRISS Iter 1 | 21.7 | 19.6 | 22.1 | 8.1 | 34.5 | 19.1 | 3.4 | 22.3 | 14.4 |
| CRISS Iter 2 | 21.8 | 20.4 | 22.6 | 11.8 | 34.6 | **19.5** | 3.9 | 22.5 | 15.5 |
| CRISS Iter 3 | **21.8** | 21 | **23.5** | **16.9** | **35.2** | 19.1 | **4.3** | 22.5 | **15.7** |

| Direction | en-lv | en-my | en-ne | en-nl | en-ro | en-si | en-tr | en-vi |
|---|---|---|---|---|---|---|---|---|
| mBART25 | **15.9** | 36.9 | 7.1 | 34.8 | 37.7 | 3.3 | 17.8 | **35.4** |
| CRISS Iter 1 | 14.1 | 36.6 | 7.8 | 34.9 | 37.6 | 3.7 | 22.4 | 35.2 |
| CRISS Iter 2 | 15.1 | 36.7 | 8.5 | 35 | 38.1 | 4 | 22.6 | 35.2 |
| CRISS Iter 3 | 15.5 | **37.2** | **8.8** | **35.9** | **38.2** | **4** | **22.6** | 35.3 |

| Direction | ar-en | et-en | fi-en | gu-en | it-en | ja-en | kk-en | ko-en | lt-en |
|---|---|---|---|---|---|---|---|---|---|
| mBART | 37.6 | **27.8** | 28.5 | 0.3 | 39.8 | **19.4** | 7.4 | **24.6** | 22.4 |
| CRISS Iter 1 | 37.6 | 26.4 | 27.7 | 8.1 | 40 | 18.7 | 9.8 | 24.1 | 22.6 |
| CRISS Iter 2 | 37.4 | 27.7 | 28.1 | 15.6 | 40 | 19 | 12.5 | 24.3 | **23.2** |
| CRISS Iter 3 | **38** | 27.6 | **28.5** | **18** | **40.4** | 18.5 | **13.2** | 24.4 | 23 |

| Direction | lv-en | my-en | ne-en | nl-en | ro-en | si-en | tr-en | vi-en |
|---|---|---|---|---|---|---|---|---|
| mBART | 19.3 | 28.3 | 14.5 | 43.3 | 37.8 | 13.7 | 22.5 | 36.1 |
| CRISS Iter 1 | 19 | 28 | 14.5 | 43.3 | 37.4 | 14.3 | 22.3 | 36.2 |
| CRISS Iter 2 | 19.7 | 28.1 | 18 | 43.5 | 38.1 | 15.7 | **22.9** | 36.7 |
| CRISS Iter 3 | **20.1** | **28.6** | **19** | **43.5** | **38.5** | **16.2** | 22.2 | **36.9** |

Table 3: Supervised machine translation downstream task

| Direction | en-ro | ro-en |
|---|---|---|
| CRISS-EnRo Iter 1 | 30.1 | 32.2 |
| CRISS-EnRo Iter 2 | 33.9 | 35 |
| CRISS25 Iter 1 | 24.9 | 27.9 |
| CRISS25 Iter 2 | 34.1 | 36.5 |

Table 4: Unsupervised machine translation results on CRISS starting from bilingual pretrained models

| Direction | Retrieval Accuracy |
|---|---|
| CRISS-EnRo Iter 1 | 98.3 |
| CRISS-EnRo Iter 2 | 98.6 |
| CRISS25 Iter 1 | 97.8 |
| CRISS25 Iter 2 | 98.5 |

Table 5: Sentence retrieval on WMT'16 English-Romanian test set

## 6.2 Multilingual Training versus Bilingual Training

Multilingual training is known to help improve translation quality for low resource language directions. Since we only use mined pseudo-parallel data at finetuning step, every language direction is essentially low-resource. We ran experiments to confirm that finetuning multilingually help with translation quality for every direction. Here, we use mBART25 encoder outputs to mine parallel data for 24 from-English and 24 to-English directions. For the bilingual config, we finetune 48 separate models using only bilingual pseudo-parallel data for each direction. For the multilingual config, we combine all 48 directions data and finetune a single multilingual model. As we can see in Figure 5 and Figure 6 multilingual finetuning performs better in general[5] and particularly on to-English directions.

## 6.3 Number of Pivot Languages

In this section, we explore how choosing the number of language directions to mine and finetune affect the model performance. [6] found that using two target languages was enough to make the sentence embedding space aligned. Unlike their work which requires parallel training data, we are not limited by the amount of labeled data, and can mine for pseudo-parallel data in every language direction. However, our experiments show that there is limited performance gain by using more than 2 pivot languages. We report the translation quality and sentence retrieval accuracy of 1st iteration

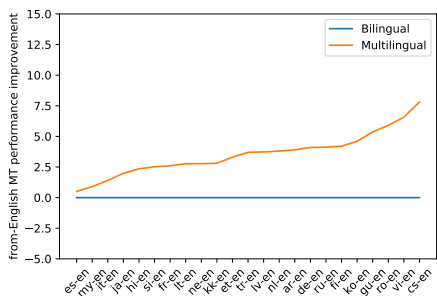

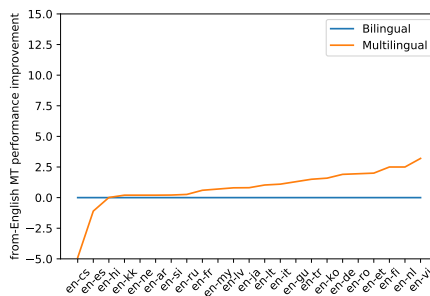

Figure 4: Bilingual versus Multingual: x-En

Figure 5: Bilingual versus Multingual: En-x

CRISS trained with 1 pivot language (English), 2 pivot languages (English, Spanish), and 4 pivot languages (English, Spanish, Hindi, Chinese).

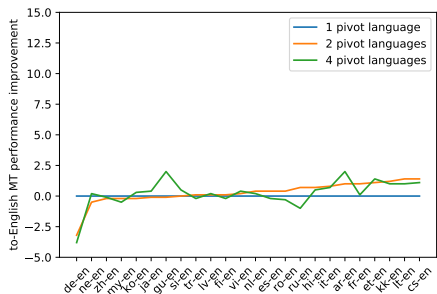

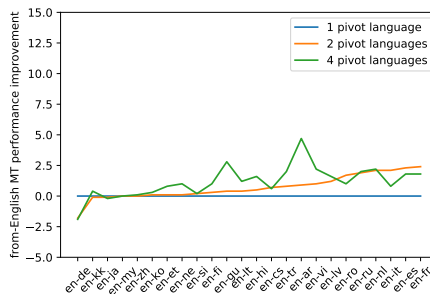

Figure 6: Pivot languages ablation: x-En MT

Figure 7: Pivot languages ablation: En-x MT

Note that the embedding space are already well aligned even with 1 pivot language because of the emergent alignment in multilingual denoising autoencoder training, and because we jointly train the from-English and to-English directions in the same model. We observe that the average translation quality and sentence retrieval accuracy improve slightly as we increase the number of pivot languages. Since using more pivot languages will increase the computational cost linearly in the mining stage, we used 4 pivot languages in the final version.

## 7 Conclusion

We introduced a new self-supervised training approach that iteratively combine mining and multilingual training procedures to achieve state-of-the-art performances in unsupervised machine translation and sentence retrieval. The proposed approach achieved these results even though we artificially limited the amount of unlabeled data we used. Future work should explore (1) a thorough analysis and theoretical understanding on how the language agnostic representation arises from denoising pretrainining, (2) whether the same approach can be extended to pretrain models for non-seq2seq applications, e.g. unsupervised structural discovery and alignment, and (3) whether the learned cross-lingual representation can be applied to other other NLP and non-NLP tasks and how.

## Broader Impact

Our work advances the state-of-the-art in unsupervised machine translation. For languages where labelled parallel data is hard to obtain, training methods that better utilize unlabeled data is key to unlocking better translation quality. This technique contributes toward the goal of removing language barriers across the world, particularly for the community speaking low resource languages. However, the goal is still far from being achieved, and more efforts from the community is needed for us to get there.

One common pitfall of mining-based techniques in machine translation systems, however, is that they tend to retrieve similar-but-not-exact matches. For example, since the terms "US" and "Canada" tends to appear in similar context, the token embedding for them could be close to each other, then at mining stage it could retrieve "I want to live in the US" as the translation instead of "I want to live in Canada". If the translation is over-fitted to these mined data, it could repeat the same mistake. We advise practitioners who apply mining-based techniques in production translation systems to be aware of this issue.

More broadly the monolingual pretraining method could heavily be influenced by the crawled data. We will need to carefully study the properties of the trained models and how they response to data bias such as profanity. In general, we need to further study historical biases and possible malicious data pollution attacks in the crawled data to avoid undesired behaviors of the learned models.

## Acknowledgments and Disclosure of Funding

We thank Angela Fan, Vishrav Chaudhary, Don Husa, Alexis Conneau, and Veselin Stoyanov for their valuable and constructive suggestions during the planning and development of this research work.

## Footnotes

[1] https://github.com/pytorch/fairseq/blob/master/examples/criss

[2]https://github.com/pytorch/fairseq/blob/master/examples/mbart/README.md

[3]We filter the test split to samples that have translations for all 15 languages: Arabic, Czech, German, English, Spanish, French, Italian, Japanese, Korean, Dutch, Romanian, Russian, Turkish, Vietnamese, Chinese (simplified). As a result, we get a dataset of 2253 sentences which are translated into 15 different languages (a total of 33, 795 sentences).

[4]https://github.com/moses-smt/mosesdecoder/blob/master/ scripts/generic/multi-bleu.perl

[5]en-cs direction is an outlier which is worth further investigation.

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
