[Supplementary Material]

# Cross-lingual Retrieval for Iterative Self-Supervised Training (supplementary materials)

## 1 Experiment details

In this section, we describe our experimental procedures in more details including hyperparameters, and intermediate results. Because of the file size limit, we will release the source code and pretrained checkpoints after the anonymity period.

### 1.1 Preprocessing details

To be able to make a fair comparison, we followed the same preprocessing steps as described in [13]. We use the same sentence-piece model [10] used in [7] and [13], which is learned on the full Common Crawl data [18] with $250,000$ subword tokens. We apply the BPE model directly on raw text for all languages without additional processing.

### 1.2 Mining details

In each iteration, we mine all 90 language pairs in parallel, using 8 GPUs for each pair, each pair taking about $15 - 30$ hours to finish. We lightly tune the margin score threshold using validation BLEU (using threshold score between 1.04 and 1.07.) We noticed small variations between different score thresholds in sentence retrieval accuracy and translation quality. The mined bi-text size for English-centric directions at each iteration are reported in Figure 1

### 1.3 Multilingual training details

For all experiments, we use Transformer with 12 layers of encoder and 12 layers of decoder with model dimension of 1024 on 16 heads ($\sim$ 680M parameters). [1] We trained for maximum $20,000$ steps using label-smoothed cross-entropy loss with 0.2 label smoothing, 0.3 dropout, 2500 warm-up steps. We sweep for the best maximum learning rate using validation BLEUs, arriving at learning rate of

Figure 1: Mined bi-text sizes for English-centric directions at each iteration

5e−5 for iteration 1, 3e−4 for iteration 2, 5e−5 for iteration 3. For all iterations, we train on 16 GPUs using batches of 1024 tokens per GPU.

## 1.4 Evaluation details

For unsupervised machine translation task, we evaluate BLEU scores using multi-bleu.perl[2] to be comparable with previous literature [8], [17], [12], [16] . For supervised machine translation task, we evaluate BLEU scores using sacreBLEU to be comparable with [13]. For both tasks, we compute the BLEU scores over tokenized text for both the reference text and system outputs. We refer readers to [13] for a detailed list of the tokenizers used.

# 2 Supplemented Tables and Figures

## 2.1 Monolingual corpus statistics

In Table 1, we report the statistics of the monolingual data we used for mining stage in CRISS.

## 2.2 Extra unsupervised machine translation results

In Table 3, we report unsupervised machine translation results for language pairs that do not have previous benchmarks. These results are generated using the

| Language code | Language name | Data size | Number of sentences |
|---|:---:|:---:|:---:|
| ar | Arabic | 20G | 100M |
| cs | Czech | 9.6G | 100M |
| de | German | 11G | 100M |
| en | English | 9.3G | 100M |
| es | Spanish | 13G | 100M |
| et | Estonian | 2.1G | 21.8M |
| fi | Finnish | 9.8G | 100M |
| fr | French | 11G | 100M |
| gu | Gujarati | 747M | 3.9M |
| hi | Hindi | 11G | 47M |
| it | Italian | 13G | 100M |
| ja | Japanese | 8.6G | 100M |
| kk | Kazakh | 1.3G | 7.5M |
| ko | Korean | 8.7G | 100M |
| lt | Lithuanian | 4.2G | 38.5M |
| lv | Latvian | 2.1G | 18.6M |
| my | Burmese | 972M | 3.3M |
| ne | Nepali | 1.2G | 4.8M |
| nl | Dutch | 9.9G | 100M |
| ro | Romanian | 14G | 100M |
| ru | Russian | 19G | 100M |
| si | Sinhala | 3.5G | 20M |
| tr | Turkish | 9.2G | 100M |
| vi | Vietnamese | 11G | 100M |
| zh | Chinese (Simplified) | 9.9G | 100M |

Table 1: Statstics of monolingual data used for mining

same models that were described in section 5.1

## 2.3 Supervised Machine Translation data source

We use the same supervised machine translation data as described in [13]

| Language code | Language name | Data source |
|---|---|---|
| ar | Arabic | IWSLT17 [5] |
| cs | Czech | WMT17 [15] |
| es | Spanish | WMT13 [2] |
| et | Estonian | WMT18 [4] |
| fi | Finnish | WMT17 [15] |
| gu | Gujarati | WMT19 [1] |
| hi | Hindi | ITTB [11] |
| it | Italian | IWSLT17 [5] |
| ja | Japanese | IWSLT17 [5] |
| kk | Kazakh | WMT19 [1] |
| ko | Korean | IWSLT17 [5] |
| lt | Lithuanian | WMT19 [1] |
| lv | Latvian | WMT17 [15] |
| my | Burmese | WAT19 [14] |
| nl | Dutch | IWSLT17 [5] |
| ru | Russian | WMT16 [3] |
| tr | Turkish | WMT17 [15] |
| vi | Vietnamese | IWSLT15 [6] |

Table 2: Test set used for unsupervised machine translation

| Direction | en-ar | ar-en | en-cs | cs-en | en-es | es-en | en-et | et-en | en-fi | fi-en |
|---|---|---|---|---|---|---|---|---|---|---|
| CRISS Iter 1 | 7.9 | 20.6 | 11.1 | 19.0 | 26.4 | 27.6 | 11.4 | 17.6 | 12.2 | 17.3 |
| CRISS Iter 2 | 13.9 | 27.0 | 16.4 | 26.4 | 32.5 | 33.4 | 16.5 | 24.2 | 19.0 | 25.3 |
| CRISS Iter 3 | 16.1 | 28.2 | 17.9 | 26.8 | 33.2 | 33.5 | 16.8 | 25.0 | 20.2 | 26.7 |

| Direction | en-gu | gu-en | en-hi | hi-en | en-it | it-en | en-ja | ja-en | en-kk | kk-en |
|---|---|---|---|---|---|---|---|---|---|---|
| CRISS Iter 1 | 9.7 | 11.2 | 9.2 | 13.6 | 21.1 | 27.2 | 4.9 | 4.6 | 2.9 | 7.4 |
| CRISS Iter 2 | 19.2 | 22.2 | 17.4 | 22.5 | 29.1 | 32.0 | 9.9 | 8.7 | 6.8 | 16.1 |
| CRISS Iter 3 | 22.8 | 23.7 | 19.4 | 23.6 | 29.4 | 32.7 | 10.9 | 8.8 | 6.7 | 14.5 |

| Direction | en-ko | ko-en | en-lt | lt-en | en-lv | lv-en | en-my | my-en | en-nl | nl-en |
|---|---|---|---|---|---|---|---|---|---|---|
| CRISS Iter 1 | 5.5 | 9.3 | 9.2 | 15.0 | 10.0 | 13.3 | 3.8 | 2.5 | 22.0 | 28.7 |
| CRISS Iter 2 | 12.8 | 15.1 | 14.4 | 21.2 | 13.6 | 18.6 | 9.5 | 4.9 | 29.0 | 34.0 |
| CRISS Iter 3 | 14.0 | 15.4 | 15.2 | 20.8 | 14.4 | 19.2 | 10.4 | 7.0 | 30.0 | 34.8 |

| Direction | en-ru | ru-en | en-tr | tr-en | en-vi | vi-en |
|---|---|---|---|---|---|---|
| CRISS Iter 1 | 13.4 | 20.0 | 9.8 | 10.8 | 21.0 | 24.7 |
| CRISS Iter 2 | 21.5 | 27.6 | 15.9 | 19.1 | 29.6 | 29.9 |
| CRISS Iter 3 | 22.2 | 28.1 | 17.4 | 20.6 | 30.4 | 30.3 |

Table 3: Unsupervised machine translation results on language directions without previous benchmarks. Refer to Table 2 for the test data source used for these language pairs.

| Language code | Language name | Data source | Number of sentence pairs |
|---|---|---|---|
| ar | Arabic | IWSLT17 [5] | 250K |
| et | Estonian | WMT18 [4] | 1.94M |
| fi | Finnish | WMT17 [15] | 2.66M |
| gu | Gujarati | WMT19 [1] | 10K |
| hi | Hindi | ITTB [11] | 1.56M |
| it | Italian | IWSLT17 [5] | 250K |
| ja | Japanese | IWSLT17 [5] | 223K |
| kk | Kazakh | WMT19 [1] | 91K |
| ko | Korean | IWSLT17 [5] | 230K |
| lt | Lithuanian | WMT19 [1] | 2.11M |
| lv | Latvian | WMT17 [15] | 4.50M |
| my | Burmese | WAT19 [14] | 259K |
| ne | Nepali | FLoRes [9] | 564K |
| nl | Dutch | IWSLT17 [5] | 237K |
| ro | Romanian | WMT16 [3] | 608K |
| si | Sinhala | FLoRes [9] | 647K |
| tr | Turkish | WMT17 [15] | 207K |
| vi | Vietnamese | IWSLT15 [6] | 133K |

Table 4: Statistics of data used in supervised machine translation downstream task

## Footnotes

[1] We include an additional layer-normalization layer on top of both the encoder and decoder, which we found stabilized training at FP16 precision.

[2]https://github.com/moses-smt/mosesdecoder/blob/master/scripts/generic/multi-bleu.perl