[Reviews · NeurIPS 2020]

Review 1

Summary and Contributions: This work combines together two lines of work on (i) parallel sentence mining using cross-lingual embeddings and (ii) Unsupervised machine translation, to come up with a novel iterative self-supervised approach to improve on both tasks. The approach involves training a multilingual de-noising auto-encoder on monolingual data from multiple languages. The representations learnt on this task are used to initialize an iterative process of mining parallel data, and then using this parallel data to fine-tune the multilingual auto-encoder on a supervised translation task. Training on supervised translation improves the alignment of the cross-lingual embeddings improving the quality of the next set of mined data, while mining better parallel data improves the quality of the model on translation, resulting in a positive feedback loop that benefits both tasks. Key Contributions: (i) The paper proposes a novel approach for unsupervised parallel corpus mining and unsupervised machine translation, improving on the SoTA on both tasks by significant margins. Experiments are conducted on the Tatoeba retrieval task and a 25 language translation task based on a combination of a few academic benchmark datasets. (ii) Careful experiments to demonstrate how using parallel data from just one language pair significantly improves the cross-lingual embedding alignment in a multilingual de-noising auto-encoder.

Strengths: This is a very interesting project combining together two lines of research, resulting in strong improvements on both tasks. (i) Solid quality improvements over strong baselines on both unsupervised retrieval and translation tasks. (ii) Careful case studies with retrieval on the 58-language Ted dataset to demonstrate how fine-tuning even on a single language pair significantly improves the quality of retrieval on all language pairs. This example clearly motivates the rest of the study. (iii) Very well motivated and easy to read.

Weaknesses: No major weaknesses. (i) The paper has several typos and grammatical errors. Line 26: Retrievial -> Retrieval Line 99 -> agnostic -> agnosticity, representation -> representations Line 152 prepossessing -> pre-processing

Correctness: No issues with claims/methods.

Clarity: Paper is usually well written and easy to read except for grammatical/spelling errors.

Relation to Prior Work: Discussion of embeddings based parallel corpus mining methods is missing from the Related work.

Reproducibility: Yes

Additional Feedback: 1. Additional experiments with varying number of languages would help understand whether the quality of mining and translation improve with increasing multilinguality. 2. mBart is a relatively large model architecture with 680M parameters. It would be interesting to see how quality on both tasks is affected when the model capacity is restricted.


Review 2

Summary and Contributions: The paper produces convincing results in supervised and unsupervised machine translation and sentence retrieval through an approach with cross-lingual retrieval for iterative self-supervision, whereby mining and training are applied iteratively.

Strengths: 1. Convincing results across three tasks. 2. Mostly thorough experimental exposition. 3. Nice case study and model description.

Weaknesses: 1. The writeup needs to be improved, e.g. multiple singular-plural mismatches. 2. The ablation study with pivot languages is not conclusive. Why is the 4-pivots curve so jittered?

Correctness: Mostly correct.

Clarity: Mostly well written.

Relation to Prior Work: Good account of related work.

Reproducibility: Yes

Additional Feedback:


Review 3

Summary and Contributions: ======= Update after the author response ======= I would like to thank the authors for the time they took to provide answers and clarifications to all my questions - and there were quite a few questions there! Overall, if the authors incorporate these responses and clarifications into the revised paper, that would make the paper much stronger, so I would strongly suggest the authors to do so. After checking the author feedback, I am more convinced that this paper does make a solid contribution to the field of unsupervised and weakly supervised MT and cross-lingual applications, and the insight on improving bilingual sentence mining in an unseen language pair by relying on sentences from another language pair is particularly interesting. All in all, I have adjusted my scores according to this reassessment of the work. In this work, the authors propose an iterative method on top of the mBART model pretrained on 25 languages. The main idea of the work is quite simple: use a module for mining additional pseudo-parallel corpora from monolingual data to continue training and tuning the initial mBART model on the pseudo-parallel corpora. The pool of pseudo-parallel corpora is augmented and refined over time which consequently leads to improvements in the mBART-based MT model. The actual "twist" is that everything is done with mBART: the unsupervised parallel data mining module is also based on fine-tuned mBART, which keeps improving both mining and MT iteratively. One interesting aspect, empirically verified in the work and which motivated the whole idea is that supplementing the model with parallel data in only a (reasonably small) subset of language pairs yields improved performance also to "unseen" language pairs. This implies that some inter-language structural similarity is implicitly leveraged by the model, and I find this a very interesting empirical finding. The initial mBART itself is a very strong starting point, and the authors show that applying the idea of iterative self-supervised training to mBART can yield even stronger performance and state-of-the-art performance almost across the board in a range of MT setups. As a sanity check, prior to MT evaluation, the authors also show the boosts in cross-lingual sentence retrieval experiments, which is a quantitative validation of the previously mentioned phenomenon of structural similarity in multilingual models such as mBART. Overall, while the results are quite strong (which can be very much credited to mBART and large-scale pretraining), the idea of iterative self-learning on similar models has been tried out before, and I do not find the paper very novel methodologically. For instance, a recent work of Artetxe et al. (https://www.aclweb.org/anthology/P19-1494.pdf) applied a very similar idea in the context of learning iteratively better and better static cross-lingual word embeddings from an unsupervised NMT model which can then, iteratively, improve phrase tables of the NMT models and yield a better NMT model. Conceptually, this is exactly the same idea as the one presented in this work (while the previous work has not been credited).

Strengths: - A simple and effective iterative self-learning idea which combines iteratively improving pseudo-parallel corpora mining with iteratively improving NMT, based on the state-of-the-art mBART model. - Very strong results in sentence retrieval and MT experiments in a range of setups (both unsupervised and supervised ones). - An interesting and empirically verified observation that fine-tuning mBART on only one language pair (i.e., their corresponding parallel corpora) can improve retrieval performance also for other (unseen) language pairs. - The paper is very easy to read and follow.

Weaknesses: - The presented iterative framework is mostly reapplying the existing ideas on pseudo-parallel sentence mining to the latest and exciting multilingual model: mBART. Therefore, methodologically, the paper does not bring anything new, and seems a bit as teaching a new dog an old trick. - The paper still lacks a lot of meaningful analysis: (i) decomposition of factors (scale of pretraining [data, languages], # sentences / language, interaction effects) required to achieve transfer capability. - In theory, the same idea could be applied to other prominent pretrained multilingual models such XLM-R, MASS, etc. The paper does not provide any insight how much of the performance gains is due to a strong starting point, and how much is due to the proposed iterative scheme per se. What would be the scores of CRISS-style tuning applied on MASS or XLM-R? Overall, the paper should provide additional experiments and insights that would profile the main properties of the method and indicate if the same methodology can be directly applied to other initial pretrained models, to bilingual setups, and with other pseudo-parallel mining mechanisms. - The method inherits the limitations of the initial mBART model: it can be successfully applied only to the 25 languages covered by mBART-25. What about all other languages? The paper should discuss how to adapt this to other truly low-resource languages - the problem is far from solved, and the paper, although claiming it in its 'broad impact statement', does not make a step forward here, due to the inherited limitations of mBART. Disclaimer: This is not a weakness in the strict sense, but rather a subjective opinion of this reviewer: I feel like I haven't learned much from this paper (and this sentiment holds after rereading it multiple times): it is mostly the application of an already existing idea to a new strong model with strong results, and as such it doesn't offer any truly novel insight or exceptional result (except for the expected SOTA performance, given that it starts from a strong model and simply improves over it with data augmentation and additional fine-tuning). Therefore, I see this more as an engineering contribution rather than as a true scientific contribution required for a conference such as NeurIPS.

Correctness: The experiments presented in the paper are sound and well executed and offer strong results. However, as suggested under "Additional Feedback" and "Weaknesses" there are additional experiments which would make the paper much stronger and would offer additional important substance. This is not a weakness of this work per se, but rather a weakness of the entire research thread on unsupervised MT: are evaluations on EN-DE and EN-FR really meaningful and useful for unsupervised models whose primary purpose should be improving performance on low-resource languages (in that respect, evaluating on the Flores datasets is definitely fine).

Clarity: Well written and easy to read. The structure of the paper is quite natural and easy to grasp. Some parts (e.g., the discussion on (not using) backtranslation) have been downplayed in the paper without proper justification.

Relation to Prior Work: The work of Artetxe et al. (ACL 2019) on iterating between learning cross-lingual word embeddings and learning better unsupervised MT models should be mentioned. A recent positional paper from Artetxe et al. (ACL 2020) on the current limitations and fallacies of unsupervised cross-lingual representation learning should also be discussed. Other recent supervised and unsupervised methods for pseudo-parallel mining should be cited and discussed, ideally also tested with mBART-25, see the following work: - https://www.aclweb.org/anthology/P19-1118.pdf - https://arxiv.org/pdf/1807.11906.pdf

Reproducibility: Yes

Additional Feedback: While it is interesting to have pseudo-parallel corpora mining and NMT integrated into a single iterative framework, the two components could and should be also tested separately. What would happen if, after iterative training as proposed, in the very last step, we combine mBART with LASER-mined parallel data (which provides the strongest performance in retrieval experiments). While LASER is a supervised method, there are parallel corpora in many language pairs now (see the position paper from Artetxe et al., ACL 2020 on false assumptions for unsupervised cross-lingual NLP work) such as WikiMatrix and JW300. If the ultimate goal is to have the strongest possible MT system, this should be a very valid experiment to try out. The authors also do not use backtranslation - they just mention it briefly in passing (Page 2). However, it would be extremely interesting, as one of the latest experiments, to combine the proposed method with iterative backtranslation to check if any gains will be met with this more expensive but potentially more effective method. I was surprised to see that such an experiment was omitted from the paper. I would also like to understand more about the benefits of the procedure for particular language pairs: e.g., I'd like to see more work and experimentation in bilingual settings focusing on a single language pair. Instead of starting from a multilingual model, can we start from, say, a bilingually pretrained model (e.g., mBART-2 pretrained only for EN-RO) and apply the same methodology? Will we see even stronger improvements or not? This would be quite instructive for the reader.

[Author Response · NeurIPS 2020]

We thank all reviewers for their thorough reviews and insightful feedback! We are encouraged that they found our work to be a novel [R1], but simple and effective [R4] way to combine two different lines of research on parallel sentence mining and unsupervised machine translation [R1]. We also appreciate that all reviewers found our work well-motivated by an interesting empirical case study [R1, R3, R4], and showed strong results by improving SoTA by significant margins [R1, R3, R4]. We address reviewer comments below and will incorporate all feedback in the final version.

**[R4] Novelty compared to [Artetxe et al 2019]**  First, we thank the reviewer for pointing us to this related work, we will gladly add a reference and discuss it in the final version. However, we would like to clarify how our work is different from it: (1) [Artetxe et al 2019] used cross-lingual word embeddings to build a phrase-based statistical machine translation system, while we use cross-lingual sentence representations to build a neural machine translation system. Therefore, our work is evaluated on tasks such as sentence retrieval, and machine translation, instead of bilingual lexicon induction. (2) Our approach shares the same neural networks architecture for pretraining and downstream tasks, making it easier to finetune for downstream tasks such as mining and translation.

**[R4] Novelty compared to other pseudo-parallel sentence mining work**  CRISS differs from existing pseudo-parallel sentence mining approaches on three important aspects: (1) Compared to supervised approaches such as LASER and [Guo et al 2018], CRISS performs mining with unsupervised sentence representations pretrained from large monolingual data. This enables us to achieve good sentence retrieval performance on very low resource languages such as Kazakh, Nepali, Sinhala, Gujarati. (2) Compared to [Hangya et. al. 2019], we used full sentence representations instead of segment detection through unsupervised word representations. This enabled us to get stronger machine translation results (37.1 BLEU vs 13.07 BLEU on WMT16 de-en). (3) Our case study demonstrated that fine-tuning even on a single language pair significantly improves the quality of retrieval on all language pairs. As mentioned by R1, this is an important new empirical finding that enabled us to iteratively self-improve the model for both mining and translation. We will add an additional related work subsection to discuss the above mentioned methods.

**[R4] Comparison to mBART as a strong starting point**  While we agree that mBART is a strong starting point, all of our results in unsupervised machine translation, sentence retrieval, and supervised machine translation are compared to mBART itself (as well as other pretraining techniques). We also included results after each iteration to show the quality improving after each step, so we believe we showed clear benefits from the iterative mining-training procedure.

**[R4] Applying CRISS-style finetuning on other pretraining techniques**  We agree that CRISS-style finetuning can be applied to other pretraining techniques such as XLM-R/MASS, and we welcome future work in this area. For this paper, we chose to start with mBART since it compared favorably with other methods on machine translation downstream tasks as well as due to page limit.

**[R4] Limit in the number of languages**  We agree that translation for low-resource languages is far from solved, and will clarify in the broader impact section that even though this work contributes to low-resource language translation, more efforts are needed by the community. CRISS' contribution to low resource translation is exemplified by our experiments on 25 languages used in mBART which contains low resource languages such as Nepali and Sinhala in Table 1 and Table 3. We will continue to explore more languages in our future work.

**[R4] Evaluation of unsupervised machine translation**  We fully agree with the reviewer that unsupervised machine translation should be evaluated on low-resource languages. We included results on En-De and En-Fr so that we can make a fair comparison with previous work on unsupervised machine translation, but we also reported results on many low-resource languages, such as the Flores test set (Ne, Si) (Table 1), and WMT 2019 (Gu, Kk) (Table 3 of supplementary materials)

**[R4] Starting with bilingual pretrained mBART**  We agree with the reviewer that the results of training CRISS starting from mBART-2 En-Ro would be instructive for the reader. We will include this experiment in the final version.

**[R1, R4] Additional ablation studies on number of languages and scale**  We had ablation studies comparing bilingual finetuning versus multilingual finetuning (Figure 4,5), and comparing between different numbers of pivot languages (Figure 6,7). In the final version, we will also include an additional ablation study on how the size of monolingual data used in mining affects unsupervised machine translation performance.

**[R4] Combination with backtranslation**  We tried finetuning CRISS further using backtranslation, but weren't able to achieve better performance. We conjecture that the mined data generated from previous iterations made the additional backtranslation data somewhat redundant/less effective.

[Meta-Review · NeurIPS 2020]

The paper proposes a novel approach for unsupervised parallel corpus mining and unsupervised machine translation, improving on the SoTA on both tasks by significant margins. Experiments are conducted on the Tatoeba retrieval task and a 25 language translation task based on a combination of a few academic benchmark datasets. Careful experiments to demonstrate how using parallel data from just one language pair significantly improves the cross-lingual embedding alignment in a multilingual de-noising auto-encoder. All reviewers support acceptance, as does the AC. Please make sure to incorporate the clarifications from the author response in the final version of the paper.